# Analysis of the Effect of Processing Conditions on Physical Properties of Thermally Set Cellulose Hydrogels

**DOI:** 10.3390/ma12071066

**Published:** 2019-04-01

**Authors:** Tim Huber, Sean Feast, Simone Dimartino, Wanwen Cen, Conan Fee

**Affiliations:** 1Department of Chemical and Process Engineering and Biomolecular Interaction Centre, University of Canterbury, Private Bag 4800, Christchurch 8020, New Zealand; sean.feast@pg.canterbury.ac.nz (S.F.); simone.dimartino@ed.ac.uk (S.D.); conan.fee@canterbury.ac.nz (C.F.); 2School of Product Design, University of Canterbury, Private Bag 4800, Christchurch 8020, New Zealand; 3Institute for Bioengineering, School of Engineering, University of Edinburgh, Edinburgh EH9 3FB, UK; 4Graymont Limited, New Zealand Region Office and Otorohanga Plant, 498 Old Te Kuiti Road RD 6, Otorohanga 3900, New Zealand; wcen603@aucklanduni.ac.nz

**Keywords:** cellulose, hydrogel, physical cross-linking, chemical cross-linking

## Abstract

Cellulose-based hydrogels were prepared by dissolving cellulose in aqueous sodium hydroxide (NaOH)/urea solutions and casting it into complex shapes by the use of sacrificial templates followed by thermal gelation of the solution. Both the gelling temperatures used (40–80 °C), as well as the method of heating by either induction in the form of a water bath and hot press or radiation by microwaves could be shown to have a significant effect on the compressive strength and modulus of the prepared hydrogels. Lower gelling temperatures and shorter heating times were found to result in stronger and stiffer gels. Both the effect of physical cross-linking via the introduction of additional non-dissolving cellulosic material, as well as chemical cross-linking by the introduction of epichlorohydrin (ECH), and a combination of both applied during the gelation process could be shown to affect both the mechanical properties and microstructure of the hydrogels. The added cellulose acts as a physical-cross-linking agent strengthening the hydrogen-bond network as well as a reinforcing phase improving the mechanical properties. However, chemical cross-linking of an unreinforced gel leads to unfavourable bonding and cellulose network formation, resulting in drastically increased pore sizes and reduced mechanical properties. In both cases, chemical cross-linking leads to larger internal pores.

## 1. Introduction

Hydrogels are three-dimensional polymer networks capable of trapping large amounts of water. Since their first mentioning in 1960, hydrogels have found applications in numerous areas including food and agriculture but also for tissue engineering, drug delivery, contact lenses, sensors, or purification and filtration systems [1].

The network-forming polymer can either be of synthetic or natural origin. Commonly used synthetic polymers include poly(vinyl alcohol), poly(ethylene glycol), or poly(acrylic acid). Those polymers can be photo polymerized and their mechanical properties and microstructure can be easily adjusted [1,2]. 

Polysaccharides like starch, alginate, or chitosan are often used to fabricate bio-based hydrogels and usually show an improved biocompatibility compared to their synthetic counterparts [1].

Cellulose has been identified as a promising bio-based polysaccharide for the manufacture of hydrogels as it provides excellent mechanical properties combined with an almost inexhaustible supply of raw materials [3]. Hydrogel beads and membranes made from regenerated cellulose have been used in numerous applications including protein immobilization or drug release [3,4,5].

Cellulose hydrogel processing usually consists of (i) dissolution of a cellulosic material in an appropriate solvent, and (ii) regeneration of the cellulose by a sol–gel transition process by the introduction of a coagulant, most often water or alcohol [6]. Available cellulose solvents include N-methylmorpholine-N-oxide (NMMO) hydrates [7], N,N-dimethylacetamide/lithium chloride (DMAc/LiCl) [8], ionic liquids (ILs) [8,9,10], or aqueous solutions containing sodium hydroxide (NaOH) or lithium hydroxide (LiOH) often in combination with urea or thiourea [11]. 

Compared to more potent solvents such as ILs or NMMO, NaOH/urea solutions are only able to dissolve relatively small amounts of up to 6–7 wt.% of cellulose [11,12]. The dissolution of cellulose becomes thermodynamically favourable at low temperatures through the formation of a cellulose–NaOH–urea “inclusion complex” (IC), where the NaOH and urea surround the cellulose chains in a hydrate layer to form a “wormlike” structure [11,12,13]. Urea is thought to help stabilise the solution by accumulating around the hydrophobic regions of the cellulose through van der Waals forces [14,15]. Interestingly, rapid gelation can be triggered at elevated temperatures (>50 °C) as a result of a thermal-induced phase separation or breakage of the ICs [16,17]. A summary of the complex dissolution behavior of cellulose in solvents based on NaOH can be found in the excellent review by Budtova and Navard [18]. This unique gelation behavior can be used to create complex three-dimensional cellulose gels using a 3D printing process [19], leading to improved performance of the gels in applications such as a stationary phase in high-performance liquid chromatography [20] or tissue engineering [21].

Gelation of the cellulose solution will depend strongly on temperature as well as the heat delivery method in the form of conduction or radiation in the form of microwaves. In particular, microwaves have been used to set a cellulose-carbonated hydroxyapatite nanocomposite [16,17,22,23]. Yet the effect of those conditions on the resulting gels is widely unexplored and will be analysed in this work to better understand the potential of those gels for applications as porous media in the form complex three-dimensional structures. 

The limited solubility of cellulose in aqueous NaOH/urea results in hydrogels with relatively low strength and stiffness, which in turn could be a limiting factor for the targeted applications [16,17,22,23,24]. Cross-linking is a common method to strengthen cellulose hydrogels and can either be based on physical interaction on a molecular scale such as ionic or hydrogen bonding or by a chemical reaction creating covalent bonds [25,26,27]. All cellulose composites use cellulose as both matrix and reinforcing phase (physical cross-linker) to produce composites with improved mechanical properties [28].

An alternative to physical cross-linking is chemical cross-linking through the formation of inter-chain covalent bonds. Epichlorohydrin (ECH) is commonly used to cross-link polysaccharides and has been employed for the cross-linking of cellulose hydrogels from NaOH/urea solutions [28,29,30,31], but can also be used in an additional post-processing step after the cellulose has been regenerated from the solvent [32]. Both physical and chemical cross-linking of cellulose in cellulose hydrogels has been analysed before, but their combined effects have not been thoroughly characterised [29,32,33,34]. 

In this work the effects of gelling conditions such as temperature and method of heating on the mechanical properties of cellulose based hydrogels prepared by thermal gelling prior to regeneration will be characterised. It will be analysed how strength and stiffness of the gels can further be improved by addition of cellulose powder acting as physical cross-linker and reinforcing agent. The effects created by combining physical cross-linking with additional chemical cross-linking through the addition of ECH, resulting in the mechanical properties compared to cross-linking in a two-step cross-linking process, will be studied. The effects of the physical and chemical cross-linking, as well as the gelling conditions on the microstructure of the hydrogels will be analysed. This work, aiming to characterize thermally set cellulose hydrogels will allow us to assess their potential for applications in the form of 3D printed porous media, specifically for use as a liquid chromatography stationary phase or 3D printing of vascular soft tissue scaffolds [35,36].

## 2. Materials and Methods

### 2.1. Chemicals

The cellulose used was Sigmacell Cellulose powder, Type 20 (average particle length 20 µm), purchased from Sigma-Aldrich (Sigma-Aldrich, St. Louis, MO, USA). Sodium Hydroxide (purity 97%) was purchased in pellet form from Thermo Fisher Scientific (Waltham, MA, USA). Urea (ACS grade), and Epichlorohydrin (ECH) (purity 98%) were also purchased from Sigma-Aldrich. All chemicals were used as-received.

### 2.2. Preparation of Cellulose Solution

A solution of 12 wt.% urea and 7 wt.% NaOH in distilled water was prepared and cooled down to −12 °C. 5 wt.% of cellulose powder was added to the solution and stirred for approximately 60 s at 300–400 rpm. Temperatures were kept at no lower than −12 °C to prevent the solution from freezing. The mixture is kept overnight at −12 °C and then stirred vigorously at 1000–1200 rpm using an overhead stirrer (IKA Eurostar overhead stirrer, IKA Works GmbH & Co. KG, Staufen, Germany) for approximately 5 min until no more cellulose particles were visible. During the complete process the solution was immersed in a bath of glycol and water (ratio 1:4) to keep the solution temperature constant at −12 °C. The solution was then centrifuged for 2 min at 3000 rpm using a MSE Centaur centrifuge (MSE Ltd., London, UK) to remove any remaining cellulose agglomerates and was then stored at 1–2 °C until further use. The final cellulose concentration was found to be 4.98 ± 0.2 wt.%.

### 2.3. Preparation of Cellulose Hydrogels

To prepare the hydrogels and hydrogel composites, approximately 40 g of the cellulose solutions were cast in transparent, square, 30 × 30 × 20 mm fully sealed Tupperware^TM^ containers and subsequently placed in a PolyScience WDS20A12E temperature controlled water bath (PolyScience, Niles, IL, USA) preheated to 40, 50, 60, 70, and 80 °C. The containers were left in the bath until the solution was gelled. Gelling was determined by tilting the containers until no flow of any remaining liquid could be detected. 

Additionally, samples were heated by using a standard 1000 W Samsung Timesaver microwave (Samsung Group, Seoul, Korea). Forty-gram samples of the dissolved cellulose solution were placed inside transparent, square, fully sealed Tupperware containers which were placed inside the microwave. The samples were microwaved at a power setting of 100 w for 30 s intervals with 4 s pauses in-between the heating intervals for a total of 15 min. At a power setting of 180 w, the pauses were extended to 30 s and the total heating time was reduced to 10 min. During each pause the samples were checked to see if gelation had occurred by a colour change as well as the solution holding together when the container was tilted on its side. Higher power setting lead to instant boiling or burning of the sample. 

Finally, to assess the effect of rapid heating by induction, a square mould with a side length of 160 mm and a thickness of 6 mm was filled with the cellulose solution. The mould was covered with square plates of stainless steel of equal side length and a thickness of 2 mm and was then placed in a hot press (Gibitre Instruments, Bergamo, Italy). The press was used to apply a pressure of approximately 0.1 MPa to ensure firm contact between the heated platens and the mould. The hot press was then set to heat to 70 °C from room temperature (~20 °C) using a heating rate of 8°/min. The mould was kept at 70 °C for 4 min to fully gel the solution and then cooled back down to room temperature at approximately 20°/min.

In all instances, the gel was removed from the mould and placed in a bath of tap water to remove the NaOH and urea and regenerate the gelled cellulose. The water was exchanged every 24 h for up to 5 days followed by additional 24 h of washing in distilled water to completely remove the NaOH and urea from the gel sheets. 

### 2.4. Preparation of All-Cellulose Composite Hydrogels

To produce all-cellulose composite hydrogels, additional cellulose powder was stirred into the solution using the overhead stirrer at approximately 800 rpm. The temperature of the solution was kept at 3–5 °C. This temperature was chosen as a compromise to avoid gelling of the solution as well as dissolution of the added cellulose powder [37,38]. The solution was stirred for approximately 5 min until the added powder was evenly dispersed in the original solution. Solutions containing an additional 10, 50, and 100 wt.% of the initially dissolved cellulose portion were prepared and gelled using the hot press method. Additionally, composite hydrogels with 50% cellulose were prepared using the water bath and microwave heating at a power setting of 100 w.

### 2.5. Chemical Cross-Linking of Cellulose Hydrogels

Chemical cross-linking was carried out either during gelation or on gelled and regenerated hydrogels. In the first case, ECH of 5, 10, or 15 wt.% of the total solution mass was added to either neat cellulose solutions or cellulose solutions containing an additional 10, 50, and 100 wt.% cellulose and stirred for 5 min at a rotational speed of 1000 rpm. In the second case, neat cellulose hydrogels or composite hydrogels containing an additional amount of 50 wt.% cellulose were prepared and weighed. An amount of 200 mL of mixtures of ECH and water (ECH concentrations of 5, 10, or 15 wt.% of the gel weight) were prepared and were heated on an IKA Yellow MAG HS hot plate to 50 °C. The gel samples were added and kept in the mixture for 2 h under light stirring. The gel samples were then removed from the water-ECH solution and rinsed with distilled water for 5 min. 

### 2.6. Gel Analysis

In total, 44 different gel samples were produced using a combination of physical, chemical, or no additional cross-linking and gelled using three different gelation methods (water bath, hot press, or microwave heating). All samples were mechanically tested, while additional testing was carried out on the gel microstructure, cross-linking efficiency, and through Fourier Transform Infrared (FTIR) Spectroscopy. A full list of all samples and analysis methods can be found in the Appendix A.

### 2.7. Mechanical Testing

Cylindrical samples (25 mm diameter) were cut from the regenerated hydrogels using a circular cutter. Testing was carried out using circular compression clamps (40 mm diameter) on a MTS Criterion Model 43 Universal Testing machine (MTS, Eden Prairie, MN, USA) equipped with a 500 N load cell at a crosshead speed of 1 mm/min. Before testing, excess water was wiped off the sample surface using paper towel, and the sample was covered with low viscosity silicon oil to avoid frictional shear during testing. The samples were then placed between the compression clamps and a pre-load of 0.1 N was applied. Samples were tested to yield. Compression testing was chosen specifically as future applications being considered for this hydrogel are liquid chromatography columns, which mainly undergo compressive forces or pressures during use. 

### 2.8. Field Emission Scanning Electron Microscopy (FE-SEM) 

Samples of approximately 2 × 2 mm were taken from the hydrogel sheets and dipped in liquid nitrogen for instant freezing. Subsequently, the samples were freeze dried using a Labconco freezone2.5 freeze drier (Labconco Corporation, Kansas, MO, USA). After drying, the samples were manually broken to create fracture surfaces for SEM analysis. The samples were gold-coated for 60 s in three cycles at 25 mA using an Emitech K975X coater (Quorum Technologies Ltd., East Grinstead, UK).

FE-SEM was performed with a JEOL 7000F FE-SEM (JEOL Ltd., Tokyo, Japan) with a probe current of 7 mA under an acceleration voltage of 5 kV.

We are aware of the possibility of ice crystal formation during freeze drying which might affect the microstructure of prepared samples [39,40]. However, comparative analysis of microstructures, specifically pore sizes, are routinely completed via freeze drying and SEM [41,42]. Future work may look at the effects of ice crystal formation on pore size either by solvent exchange or temperature gradient [39,43]. 

### 2.9. Pore Size Analysis

Pore size distribution of the different hydrogels was estimated from the SEM micrographs using ImageJ (ImageJ, v 1.44j, Wayne Rasband, National Institutes of Health, Bethesda, MA, USA). A circular pore shape was approximated and 300–500 measurements were taken for each formulation using 8–12 SEM micrographs per formulation.

### 2.10. Cross-Linking Efficiency

To compare the effectiveness of the two cross-linking methods, samples of approximately 0.5 grams were cut from the prepared hydrogels and immersed in 100 mL of NaOH-urea solvent. The samples were kept at −12 °C for 24 h to dissolve all non-cross-linked portions of the cellulose. The samples were then rinsed in distilled water and visually assessed. 

### 2.11. Fourier Transform Infrared (FTIR) Spectroscopy

FTIR was carried out using a Bruker Vertex 70 spectrometer and OPUS operating software (Bruker Optics, Lower Hutt, New Zealand). The wavenumber range scanned was 4000–630 cm^−1^; 128 scans of 4 cm^−1^ resolution were signal-averaged. A Mercury cadmium telluride (MCT) detector was used and cooled in liquid N_2_. Attenuated total reflectance (ATR) mode was used and data recorded as an absorbance spectrum. Each sample was held inside a custom-made stainless steel chamber that was connected to a Watlow series 989 temperature controller (Watlow, Christchurch, New Zealand). The samples were heated from 25 to 85 °C in 5 °C intervals at a heating rate of approximately 10 °C/min and then kept at 85 °C for 15 min to achieve full gelation and cross-linking. The samples were allowed to stabilize for 60 s before a measurement was carried out. 

### 2.12. Manufacture of Cellulose Gels with Complex Geometric Features

Designs for the sacrificial templates were constructed using Mathematica and Solidworks. A Schoen Gyroid unit cell with an edge length of Pi was created using Mathematica according to the following formula to approximate the surface [44,45]: (1)sin(x)×cos(y)+sin(y)×cos(z)+sin(z)×cos(x)=0

The unit cell was scaled to achieve channel sizes of 500 µm and then patterned using Solidworks and multiplied to create a model of a cylinder with a radius of 1 cm and a length of 5 cm. The model was saved as a stereolithography (STL) file and transferred to a Solidscape 3ZPRO 3D printer. The model was printed using water soluble Solidscape model material. After printing and removal of support material, the model was vacuum infused with a cellulose solution containing 50% added, undissolved cellulose particles. After filling of the mould it was placed in an oven at 50 °C for 2 h to gel the cellulose solution. Subsequently the filled mould was placed in a hot water bath (~90 °C) to both regenerate the cellulose portion and simultaneously remove the Solidscape model material. The water was exchanged 5 times after which the remaining cellulose gel was rinsed with acetone to remove any residual Solidscape model material. The gel was stored in MiliQ water.

## 3. Results and Discussion

### 3.1. Manufacture of Cellulose Hydrogels with Complex Structure 

Using the simple negative templating approach, a cellulose hydrogel could successfully be manufactured into a design structure (Figure 1). The features designed and printed are well maintained in the cellulose gel allowing the production of cellulose hydrogel with complex, three dimensional structures in the sub-millimetre range. 

Complex structures such as these could be used for chromatography stationary phases or as soft tissue scaffolds where a defined flow channel is beneficial [46,47].

### 3.2. Influence of Temperature, Gelling Method and Physical Cross-Linker on Mechanical Properties of Cellulose Hydrogels

The effect of gelling method and temperature on the mechanical properties of neat (non-cross-linked) hydrogels is presented in Figure 2. Gelling in a water bath revealed that both compressive modulus and strength increased as the constant gelation temperature decreased. Cellulose degradation in alkaline conditions has been observed in the Kraft pulping industry [48]. The kinetics of this reaction speeds up as the temperature increases, producing a cellulose matrix with a lower degree of polymerisation at higher temperatures [48,49], and therefore the resulting hydrogels show lower mechanical properties [50]. At the same time, the hydrolysis reaction competes with the gelation process, thus partially explaining the slight deviations from the main trend observed at 40 °C and 80 °C. The gelation time was 48 h at 40 °C, compared with 6 h at 50 °C. The extended time required for gelation at 40 °C, even if at lower temperature, might have led to greater overall amounts of degraded cellulose than those produced by alkaline hydrolysis at 50 °C, thus resulting in a weaker hydrogel. On the other hand, at 80 °C gelation occurred in only 35 min, greatly limiting the time span over which the degradation reactions could occur, eventually resulting in a slightly stronger gel than the one obtained at 70 °C and a gelation time of 65 min.

Use of a temperature ramp in the hot press with a total heating time of 10 min resulted in hydrogels with higher strength and stiffness values compared to gelation in the constant water bath at 70 °C (Figure 2). We believe this to be related to the much-reduced gelation time in the hot press compared to the water bath, and thus the strongly reduced time for any cellulose degradation to occur. The difference in gelation time between the water bath at 70 °C and hot ramp at 70 °C is due to the difference in heat transfer properties between gelation methods. The plastic container used for the water bath and subsequently larger volume of solution it contained limited this heat transfer compared to that of the hot press.

The time required for complete gelation using microwave radiation was 20 and 10 min at 100 w and 180 w, respectively, much shorter than the one required in the water bath at 50 °C (6 h) or at 60 °C (2 h). In addition, both microwave samples were no warmer than 60 °C when completely gelled. The difference in gelation times is due to more uniform and rapid heating using microwave radiations compared to the water bath, where heat is instead provided via conduction [51]. As a result of short gelation times and low temperatures, microwave gelation provided the stiffest gels, with compressive moduli of 651 ± 100 kPa and 640 ± 40 kPa at 100 w and 180 w, respectively, while strength was comparable to that achieved in a water bath at 50 °C (Figure 2). This fits the above hypothesis that faster gelling limits cellulose degradation in alkaline conditions.

Composite hydrogels were prepared adding micron sized cellulose powder on saturated cellulose solutions. Specifically, composites incorporating 10, 50, and 100 wt.% of added cellulose particles and gelled through the different heating mechanisms were considered in this work. All-cellulose composites are an interesting class of composite materials where both the matrix and reinforcement agent are of exactly the same nature, thus favouring bonding between the two phases with a consequent improvement in mechanical performances [28].

As reported, the mechanical properties of all-cellulose composites progressively and consistently increased with added content of undissolved micron-sized cellulose (Figure 2). This result is an immediate reflection of a strong bonding between the matrix and reinforcing particles as well as a good distribution of the particles throughout the gel. In particular, both the strength and stiffness of the composite gels increased, on average, by a factor of 1.15, 2.3, and 3 when adding 10, 50, or 100% microcrystalline cellulose, respectively, with respect to the neat hydrogels. Interestingly, the relative increase in mechanical properties is unaltered by the gelation method or the gelation temperature. This suggests that temperature is affecting to a similar extent both the formation of the cellulose network, as well as the creation of stable bonds between the porous matrix and reinforcing particles, hence resulting in a proportional increase in the mechanical properties of the gels investigated.

These data are consistent with the results of Wang and Chen, who observed a linear increase in shear modulus when using nano-cellulose fibres as a reinforcing agent in cellulose hydrogels produced by thermal gelation from aqueous NaOH/urea solutions [52]. 

The excellent bonding between regenerated cellulose and cellulose particles is clearly visible in Figure 3 for a gel with 100% of added cellulose. Here, in a cryo-fractured sample, an individual cellulose particle is coated with the porous regenerated matrix, and no interfacial area is visible. 

### 3.3. Properties of Chemically Cross-Linked Cellulose Hydrogels

ECH is a widely used cross-linking agent for polysaccharides [30,53]. Under basic conditions, ECH is able to link two C6 alcohol groups in cellulose, forming either intra or inter-chain cross-links. ECH cross-linking of cellulose hydrogels was produced following two alternative methods; in the first, ECH was included in the formulation of cellulose solutions, with cross-linking reactions occurring at the same time as gelation was produced. In the second, ECH treatment was applied following full gelation of the cellulose solutions. Both neat cellulose solutions and solutions with suspended cellulose were considered in these tests.

#### 3.3.1. Concurrent Chemical Cross-Linking and Gelation of Cellulose Hydrogels

In Figure 4 the mechanical properties of ECH co-cross-linked cellulose hydrogels are reported. ECH produces a change in the mechanical properties of both neat gels and gels filled with 50% additional cellulose. In particular, neat ECH co-cross-linked gels have slightly reduced stiffness and strength compared to gels obtained without ECH. We believe that due to the small amount of cellulose available for cross-linking, the majority of cross-links obtained are predominately within single, isolated cellulose chains. In turn, those intra-chain cross-linked cellulose molecules will gel in a less crystalline configuration, resulting in lower mechanical strength and stiffness. A similar behaviour was shown by Guo et al., where cross-linking small amounts of cellulose by ECH in an aqueous solution lead to a reduction in a crystallinity of the produced cellulose films and as a result lower stiffness and an overall softening of the film [54].

On the other hand, strength and stiffness increase when gels with added 50 wt.% of cellulose are treated with ECH. In this instance, cross-linking reactions occur at the same time as the gelation process, i.e., when solvated cellulose chains are still free in solution and before the gel is formed. The relatively high cellulose concentration present, originating from both the solvated cellulose and suspended cellulose particles, creates an environment rich in hydroxyl groups, thus providing a large repository of reaction sites for the attack of the epoxy groups in ECH. In particular, a substantial number of cross-links are expected to form between the free cellulose chains in solution and the dispersed cellulose fibres, resulting in a highly cross-linked cellulose network endowed with a dramatically increased strength and stiffness [34,55,56].

In addition, the kinetics of the cross-linking reactions is highly promoted at elevated temperatures, partly offsetting alkaline hydrolysis of cellulose chains. This, in turn, explains the difference in the relative increase of mechanical properties observed at the two temperatures, with a 1.5-fold increase when the composite gels are obtained at 50 °C, and a 2.5 fold increase at a gelation temperature of 70 °C. 

Interestingly, the concentration of ECH used did not produce a statistically significant change in mechanical properties in the range of 5–15% ECH (p < 0.05 for all mean values), for both neat solutions and solutions with added cellulose particles. Qin et al. reported an increase in compressive modulus of cross-linked cellulose hydrogels when increasing the amount of cross-linker from 5 to 10 and 15%. However, their cellulose solutions were made up from 3 wt.% of dissolved cellulose [34]. Furthermore, it is not clear whether their amount of added ECH was in relation to gel mass, cellulose dry mass, or any other constant, thus direct comparisons between the amount of added ECH and resulting mechanical properties cannot be drawn. 

Those observations are supported by FTIR analysis of the hydrogels produced from different formulations, reported in Figure 5 and Figure 6. Only physical cross-linking is observed for the neat hydrogel and the 50% added cellulose particles hydrogel, with main changes in the 3625–2900 cm^−1^ (Figure 5A) envelope due to the rearrangements of the hydroxyl groups and their associated hydrogen bonds during gelation. Changes in the 1625–1400 cm^−1^ region associated with the urea main vibrational modes (main peaks at 1600 cm^−1^ (δasNH_2_ and νCO) and 1450 cm^−1^ (νsCN)) can also be observed (Figure 5B) [57,58]. This is indicative of conformational changes of urea molecules, which transition from a relatively constrained state surrounding the cellulose chains (before gelation), to a more disordered state in a free solution after gelation. Finally, minor spectral changes are present in the 1150–950 cm^−1^ region (νCO, δCOC, νCC, νCOH and ring vibrations in polysaccharides), indicative of a change from preferred cellulose–solvent interaction to cellulose–cellulose interactions mediated by physical cross-links (Figure 5B) [17,59].

Samples co-cross-linked with ECH show all features observed with only physically cross-linked gels. In addition, clear vibrational changes reveal the presence of covalent bonds in the hydrogel, with the presence of new strong vibrational modes in the 1250–950 cm^−1^ region related to the formation of new ether bonds (νCOC at 1110 cm^−1^) between cellulose and ECH, and the formation of secondary alcohols in the β-hydroxypropyl ether bridges (νCO in secondary alcohols at 1030 cm^−1^). Also, the loss at 1260 cm^−1^ demonstrates a reduction in epoxy functionalities (νCO in epoxy groups), further confirming the reactivity of ECH. Finally, the blue-shift on the νOH region at 3600–3000 cm^−1^ region indicates a partial decrease in the hydrogen bonded state present in the hydrogels, related to hindered formation of hydrogen bonds in the rigid covalently cross-linked network. A change in the OH bonds is also clearly visible in the region of 3000–2800 cm^−1^. This is a typical change in cross-linked cellulose and can also be observed for polysaccharides such as starch, dextran, pullulan and carboxymethylated cellulose [60]. 

We believe the two cross-linking processes that occur during the gelling lead to the observed changes in the mechanical properties of the cross-linked samples; (i) the chemical cross-linking process caused by the ECH and described as a combination of (a) the cross-linking of the cellulose chains by β-hydroxypropyl ether bridges, (b) the substitution of OH-groups in the anhydroglucose units by dihydroxypropyl ether groups, and (ii) the reaction of OH– and ECH to glycerol occurring at approximately 50 °C [29,30] 

The two described cross-linking processes lead to observed patterns and help to explain the reduced mechanical properties of solutions with no added cellulose, as the self-cross-linking is more likely to occur when lower amounts of OH groups from cellulose are available [61]. This can be seen in Figure 6, comparing a solution with no added cellulose and 15% ECH to a solution with the same amount of ECH but 50% of additional cellulose present. A peak increase in the region around 3400 cm^−1^ for the neat solution indicates a higher concentration of hydroxides (OH) and thus pointing to a higher amount of glycerol present. This is an indication towards the reaction of excess ECH with itself resulting in the formation of glycerol instead of cross-linking bonds between cellulose [62]. 

It was also reported that a prolonged reaction time can contribute to a higher production rate of glycerol when cross-linking polysaccharides with ECH [63,64]. Additionally, with increasing gelling temperature, and with the addition of cellulose particles, this in turn creates a complex system of factors affecting the mechanical properties of the hydrogels. While lower gelling temperatures lead to less cellulose degradation and in turn higher strength and stiffness, those effects are reversed when ECH is introduced, as longer gelation times will result in higher amounts of self-cross-linking and thus lower values for compressive strength and modulus. However, with additional cellulose suspended in the solution, the ECH leads to strong covalent bonding as shown by drastically improved strength and modulus (Figure 4, Figure 7), in addition to strongly reduced gelation time, preventing degradation of the polymer chains. 

#### 3.3.2. Post Cross-Linking of Cellulose Hydrogels

Regenerated neat and all-cellulose composite hydrogels were post-cross-linked by immersion in an ECH-water mixture. In contrast to the hydrogels obtained using concurrent cross-linking, all post-cross-linked hydrogels tested showed an increase in mechanical properties (Figure 7). As previously described, this observation reflects the strengthening of the cellulose network thanks to the formation of chemical cross-links. The influence of ECH concentration is negligible also in this case, with a statistically significant increase in the mechanical properties already achieved at 5% ECH, and practically unaltered from this value at higher ECH concentrations. Similar effects have been observed for chitosan fibres where the wet tensile strength of cross-linked fibres did not increase with increasing amounts of ECH [55]. However, for the composite hydrogels with 50% added cellulose the strengthening obtained via post-cross-linking is significantly weaker (*p* < 0.05) than that achieved in the concurrently cross-linked gels. This is the result of the limited cross-linking capability of ECH on a completely formed hydrogel, characterized by cellulose chains in the network with heavily reduced mobility, thus limiting the number of inter-chain cross-links that is possible to form.

The quality of the ECH mediated cross-links produced in the two treatments, concurrent and post gelation, was checked by immersing two samples of the 50% added cellulose hydrogel in the NaOH and urea solvent. As can be observed in Figure 8, the limited presence of intermolecular cross-links obtained with post-cross-linking ECH treatment resulted in a partial sample redissolution. What remains are the outer layer of the sample, while the inner core has been dissolved, resulting in the sample collapsing. In contrast, the sample co-cross-linked did not dissolve back into the solution, confirming the strong covalently bonded network of cellulose chains throughout the whole sample. 

The produced hydrogels show a competitive range of mechanical properties compared to other similar cross-linked hydrogels, especially when cross-linked by ECH [34]. The moduli of the cross-linked gels containing additional reinforcement achieve a level of compression modulus similar and even higher compared to cellulose composite gels containing cellulose nano-whiskers [52]. The hydrogels also compare well with other polysaccharide based hydrogels such as more commonly used agarose [65,66] or hybrid gels made from hyaluronic acid and PEG [67].

### 3.4. Analysis of the Gel Microstructure

The gelling method employed did not produce a visible change in the microstructure of the hydrogels, exemplarily shown for the gels prepared without chemical or physical cross-linker in Figure 9. 

Thus, the causes for the different mechanical properties observed in Figure 4 are solely to be found at the molecular scale and related to different degradation degrees of the cellulose chains. On the other hand, a clear microstructural change is visible as a result of chemical and physical cross-linking. The addition of cellulose powder leads to a drastic reduction in average pore size (Figure 10 and Figure 11). As suggested by Wang and Chen, the additional cellulose behaves as a cross-linking agent as the additional available hydroxyl groups will result in a denser concentration of hydrogen bonds and thus overall comparatively smaller pore sizes [52,68]. This was also shown by Hoepfner et al. reporting that the presence of cellulose fibrils will lead to a denser and more stable network structure in cellulose based aero- and cryogels compared to gels without additional fibrils [69]. 

As can be seen from the pore size distribution this effect is mainly visible for the gels reinforced with 50 and 100 wt.% of cellulose (Figure 10). A small reduction in median pore size is visible for the 10 wt.% gel and the distribution is slightly skewed towards smaller pore sizes, possibly indicating the onset of pore size change but without statistical significance. However, no direct correlation between pore size distribution and mechanical properties can be observed, emphasizing the two separate effects of added cellulose powder acting simultaneously but not dependently as physical cross-linking agent and reinforcing phase. 

Employing ECH during the gelation process results in strongly increased internal pore sizes, but no direct influence of the combination of physical and chemical cross-linking can be seen (Figure 10). Interestingly, while adding 50 wt.% cellulose powder reduces the pore size approximately by a factor of 10, this effect is completely inverted, with the combination of additional cellulose powder and ECH, leading to a pore size that is approximately 10 times larger than in the untreated gel. The effect of the additives porous structure of the gel is exemplarily shown for the neat gel, a gel with 50 wt.% added cellulose powder and a gel with 10 % ECH added in (Figure 11).

It is proposed that the chemical cross-linker, in this case ECH, forms structures of large connected sheets which gives rise to pores. Due to the stronger nature of these covalent bonds during gelation larger pockets of solvent can form resulting in larger pores. Gels without chemical cross-linking display a shorter more random structure of smaller string-like pores as can be seen in Figure 3. Without the strength of the covalent bonds from chemical cross-linking, only smaller pockets of solvent are able form where cellulose break free from its IC’s as the solvent is heated. However, neither the thermal gelation process of cellulose solutions in NaOH/urea nor the processes of simultaneous physical and chemical cross-linking in the presence of thermal gelation are fully understood. Thus, future work will look to explain the interaction of those processes and their effect on this microstructure phenomenon further. In contrast, post-crosslinking leads to no significant changes of the gels internal pore sizes, supporting the above made assumptions that the cross-linker mainly acts on the gel surface at the used processing conditions. 

## 4. Conclusions

Cellulose hydrogels can be easily produced by temperature induced gelling and the used temperature and corresponding gelling time could be shown to have a strong influence on the mechanical properties of the gels. The combination of short gelling time and low gelling temperature resulted in the strongest gels. The mechanical properties of the hydrogels can be improved by the addition of cellulose powder to create stronger and stiffer all-cellulose composite hydrogels. The added cellulose acts as a physical cross-linking agent strengthening the hydrogen-bond network as well as a reinforcing phase. 

Direct chemical cross-linking via ECH leads to a significant increase in compressive gel strength and modulus in combination with physical cross-linking due to a combination of hydrogen and covalent bonds, further improving the network strength in a simple one-step process. However, chemical cross-linking of an unreinforced gel leads to unfavourable bonding and cellulose network formation, resulting in drastically increased pore sizes and reduced mechanical properties. 

Cross-linking of cellulose hydrogels using ECH subsequent to gelling and regeneration does lead to an increase in mechanical properties, but to a lesser extent than the direct cross-linking as a result of a sleeve-core effect of the cross-linking. Via negative wax templating, a complex structure of hydrogel can also be produced with defined complex channels. Mechanical strength, pore size control, and the moldable nature of the solution prior to gelation show that this hydrogel is a promising material for a 3D printable chromatography stationary phases or soft tissue scaffolding. 

## Figures and Tables

**Figure 1 materials-12-01066-f001:**
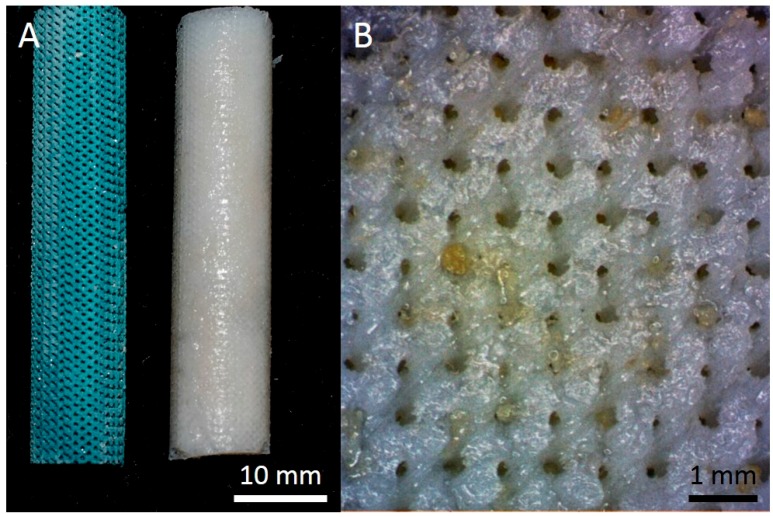
Cellulose hydrogel with a Schoen Gyroid structure made from a 3D printed mould. Shown are a side view of the printed part and corresponding hydrogel (**A**) and a cross-sectional view of the produced gel (**B**).

**Figure 2 materials-12-01066-f002:**
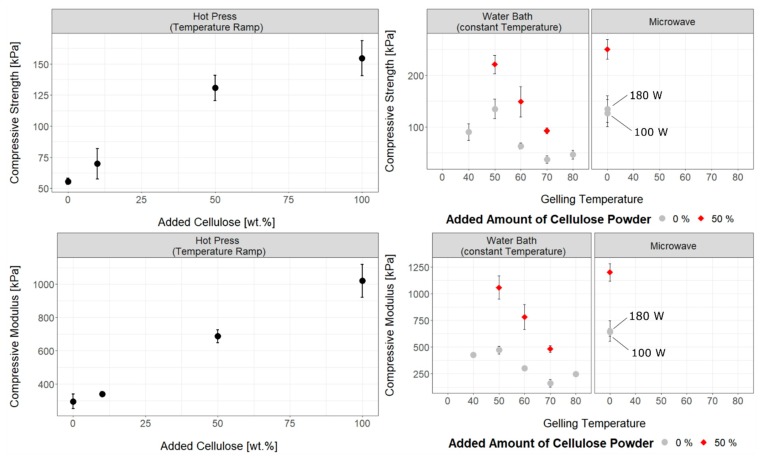
Compressive strength and modulus of the hydrogels prepared. Shown are the properties of gels prepared using different heating methods (set temperature in the water bath, temperature ramp in the hot press, and heating by microwave radiation), at different temperatures and gels reinforced by additional cellulose powder.

**Figure 3 materials-12-01066-f003:**
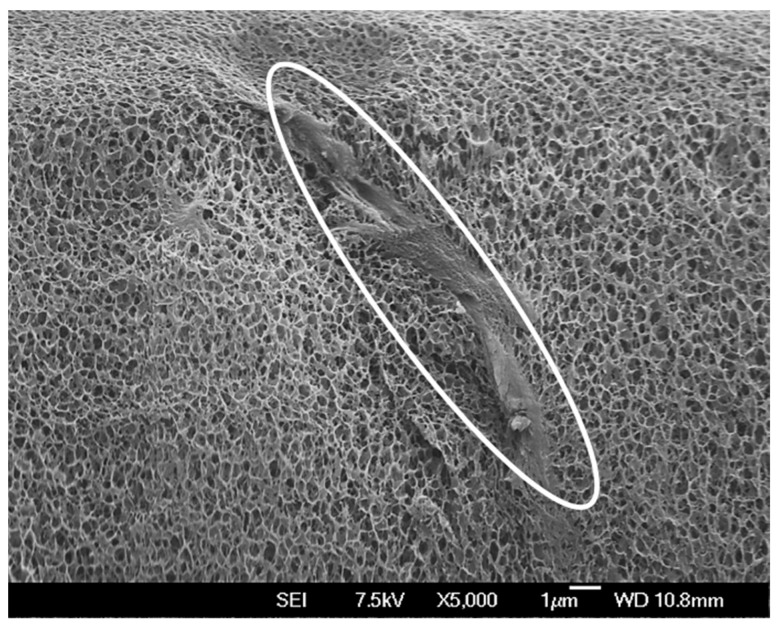
SEM micrograph of a composite hydrogel with 50% added cellulose powder. A single cellulose particle (marked by the white oval) is visible that is closely surrounded by the regenerated, porous cellulose matrix, showing excellent bonding between particle and matrix.

**Figure 4 materials-12-01066-f004:**
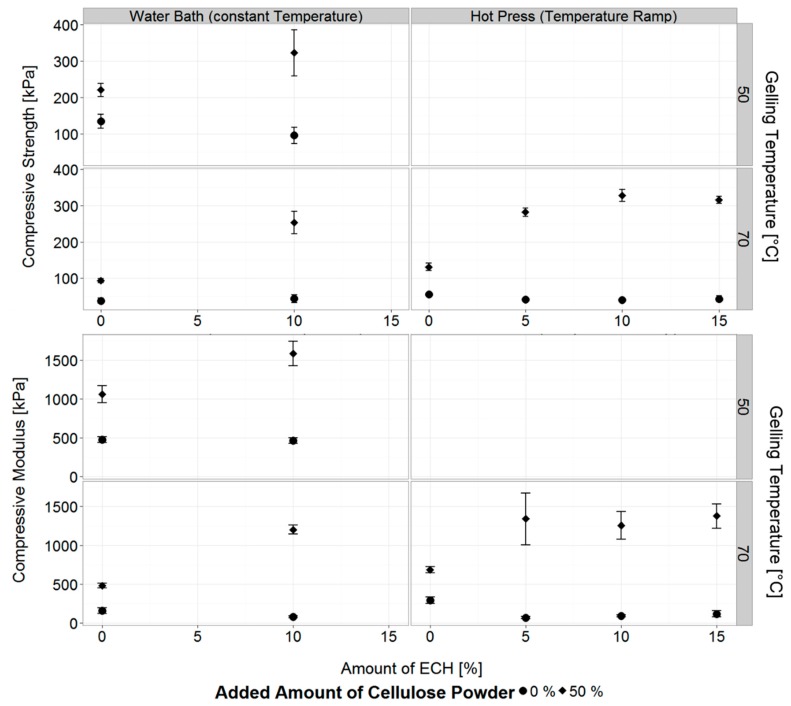
Compressive strength and modulus of cellulose hydrogels made from neat solution and solution with 50% suspended cellulose particles, cross-linked with various amounts of epichlorohydrin, and gelled at 50 and 70 °C using the water bath and hot press, respectively. Shown are mean values with one standard deviation.

**Figure 5 materials-12-01066-f005:**
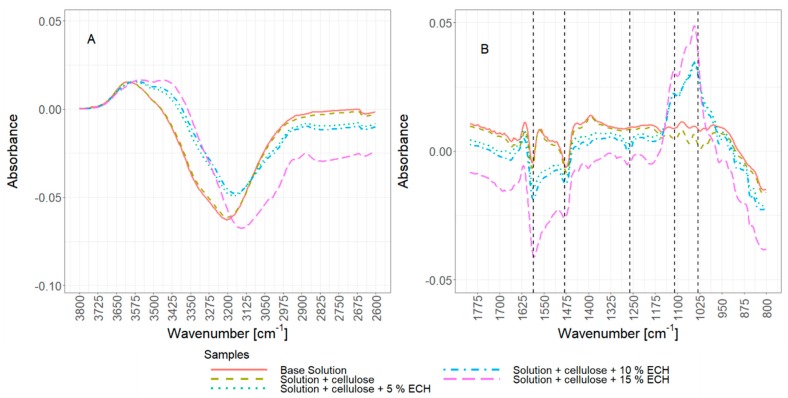
IR spectra of gelled cellulose solutions with different formulations recorded after heating the samples to 85 °C. Shown are the base solution, a solution with 50% w/w added cellulose (Solution + cellulose), and the solution with added cellulose cross-linked with 5, 10 and 15% w/w epichlorohydrin (ECH) (Solution + cellulose + 5%ECH, Solution + cellulose + 10%ECH, Solution + cellulose + 15% ECH respectively). Dashed lines mark the main peaks at 1585, 1480, 1260, 1110 and 1030 cm^−1^. Reference backgrounds: matching precursor cellulose solution before gelation at room temperature. Shown are two regions of interest from 3800–2600 cm^−1^ (**A**) and 1775–800 cm^−1^ (**B**).

**Figure 6 materials-12-01066-f006:**
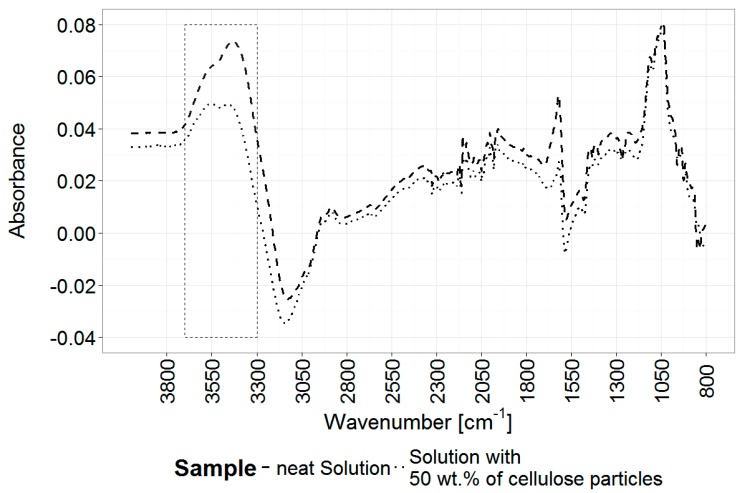
Recorded Spectra of gelled cellulose solution and solution with added cellulose powder. Shown are the spectra cross-linked in situ by the addition 15% ECH, respectively. The area of interest indicating the increased concentration of hydroxides is marked by the box. The used reference backgrounds were the respective samples at room temperature.

**Figure 7 materials-12-01066-f007:**
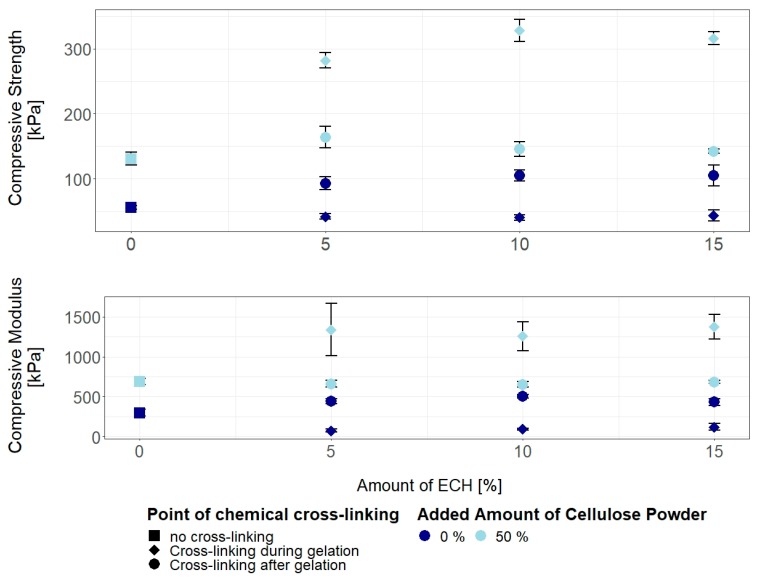
Compressive strength and modulus of cellulose hydrogels prepared by heating neat solution and solution with 50 wt.% of suspended cellulose particles, gelled in the hot press at 70 °C, and cross-linked with various amounts of epichlorohydrin either during gelation or after gelation and regeneration, compared to non-cross-linked gels. Shown are mean values with one standard deviation.

**Figure 8 materials-12-01066-f008:**
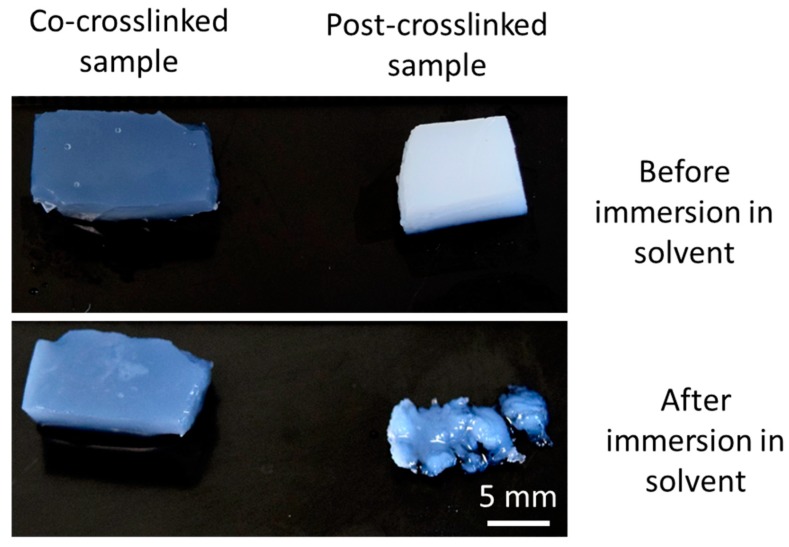
Cellulose hydrogels (50% added cellulose powder) cross-linked with 10% ECH. The sample on the left has been co-cross-linked and gelled, while the sample on the right was post-cross-linked after gelation and regeneration of the cellulose gel. Shown are the samples before and after immersion in the NaOH/urea solvent at −12 °C for 24 h. While the co-cross-linked sample remains unchanged, the post-cross-linked sample has been partially dissolved.

**Figure 9 materials-12-01066-f009:**
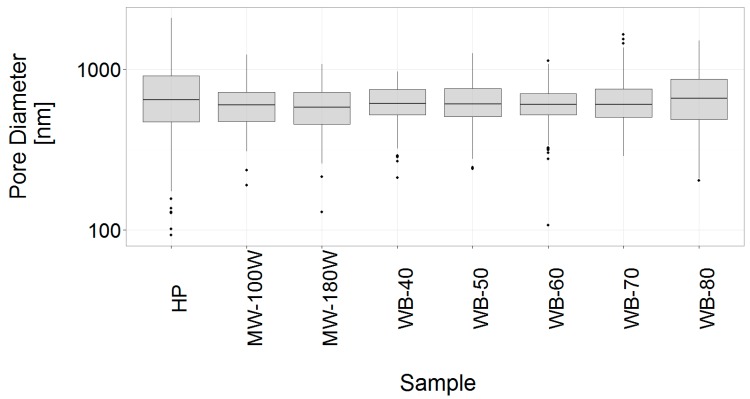
Distributions of the measured internal pore diameters of freeze-dried hydrogels with no added chemical or physical cross-linker prepared using the hot press (HP), microwave at 100 and 180 W (MW-100W & MW-180W), as well as the water set to 40, 50, 60, 70 and 80 °C (WB-40, WB-50, WB-60, WB-70 and WB-80).

**Figure 10 materials-12-01066-f010:**
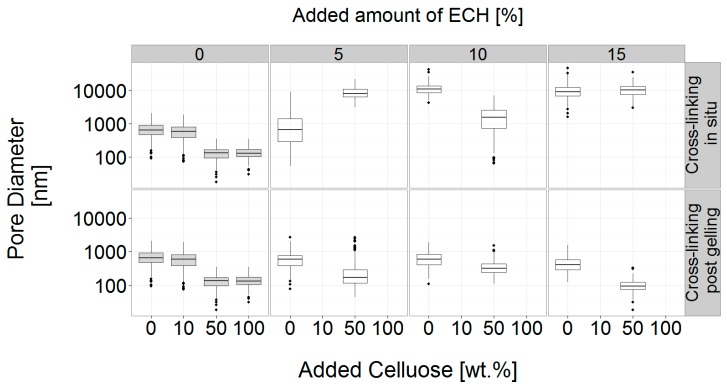
Distributions of the measured internal pore diameters of all freeze-dried hydrogel formulations prepared using the hot press.

**Figure 11 materials-12-01066-f011:**
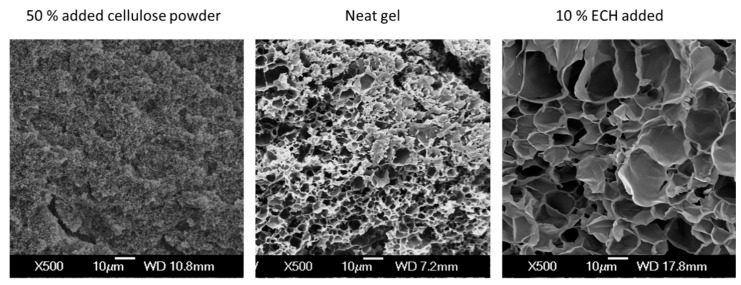
SEM micrographs showing the changes in internal porosity of the neat cellulose gel compared to the gel with 50 wt.% added cellulose powder, and the gel co-cross-linked with 10% ECH.

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
