# Peer review of "Analysis of the Effect of Processing Conditions on Physical Properties of Thermally Set Cellulose Hydrogels"

_materials, 2019, doi:10.3390/ma12071066_

Round 1
Reviewer 1 Report
Review report on "materials-468472"
This manuscript reports the mechanical property of cellulose hydrogel reinforced with several different methods such as the addition of cellulose powder, chemical cross-linking, and both of them. Although the results and discussion in this study are to be published in academic journals, the structure of the manuscript, especially paragraph writing, is too poor for the publication. Therefore, this manuscript cannot be accepted for publication in the current form and needs re-writing.
#1. grammar
Several typos are found in the main text. The authors are strongly advised to re-check the grammatical errors.
#2. paragraph
The structure of paragraphs is scattered and thus quite puzzling for the readers, especially in the Introduction. The authors should mention 1) why heat gelation is necessary, 2) why the addition of microcrystalline cellulose is necessary, 3) why chemical crosslink is necessary, and 4) why the formation of the complexed structure by the use of 3D-printing are necessary.
#3. References
Several refs in Introduction should be reconsidered (e.g. regarding Ref 14, Isobe et al. should also be cited. https://link.springer.com/article/10.1007/s10570-012-9800-7).
#4. Micro-pore structure in Fig. 11.
Micro-pore structure seen in Fig. 11 seems the result of the ice-crystal formation. The authors are advised to see the effect of ice-crystal formation by performing the solvent exchange from water to EtOH, and finally to t-BuOH as reported in the cellulose aerogels fabrication (e.g. https://onlinelibrary.wiley.com/doi/full/10.1002/anie.201405123).
Author Response
We thank the reviewer very kindly for the critical and insightful comments provided. Please find our point-by-point response below:
#1. grammar
Several typos are found in the main text. The authors are strongly advised to re-check the grammatical errors
We have had the manuscript checked by 2 additional native speakers and have corrected all found typos and mistakes. e would kindly ask the reviewer to point out any additional mistakes found including the line number, so they can be corrected if necessary.
#2. paragraph
The structure of paragraphs is scattered and thus quite puzzling for the readers, especially in the Introduction. The authors should mention 1) why heat gelation is necessary, 2) why the addition of microcrystalline cellulose is necessary, 3) why chemical crosslink is necessary, and 4) why the formation of the complexed structure by the use of 3D-printing are necessary.
We have added several sentences to the introduction explaining further the motivation for research and specified potential field of application for the created gels which should help explain the used methods of formulation and analysis.
#3. References
Several refs in Introduction should
be reconsidered (e.g. regarding Ref 14, Isobe et al. should also be
cited. https://link.springer.com/article/10.1007/s10570-012-9800-7).
We have added the specified reference and several others to the introduction.
#4. Micro-pore structure in Fig. 11.
Micro-pore structure seen in Fig. 11 seems the result of the ice-crystal
formation. The authors are advised to see the effect of ice-crystal
formation by performing the solvent exchange from water to EtOH, and
finally to t-BuOH as reported in the cellulose aerogels fabrication (e.g. https://onlinelibrary.wiley.com/doi/full/10.1002/anie.201405123).
We appreciate the possibility of ice crystal formation during freeze drying which might affect the microstructure of prepared sample. However comparative analysis of microstructures specifically pore size are routinely completed via freeze drying and SEM. This has been clarified in section 2.8 and corresponding references have been added. Future work may look at the effects of ice crystal formation on pore size either by solvent exchange or temperature gradient. We are also investigating using mercury intrusion porosimetry for future work to analyze the gels in their hydrated state.
Reviewer 2 Report
The English should be improved. For example, in line 101, "will be also be analysed" is obviously not correct.
For the compressive tests, the stress-strain curves should be presented.
For mechanical testing, why did the authors choose compressive tests instead of elongation tests?
A schematic illustration of the chemical structures and the hydrogel formation would be very helpful for making the discussion clear.
Author Response
We thank the reviewer very kindly for his constructive and informative comments.
The English should be improved. For example, in line 101, "will be also be analysed" is obviously not correct.
The manuscript has been cross-checked by several English native speakers. The mentioed typo has been corrected.
For the compressive tests, the stress-strain curves should be presented.
A large amount of samples were tested so providing the stress strain curves for them would require several more figures but provide little additional information. However, we will make the collected data available online and as supplementary file with the manuscript.
For mechanical testing, why did the authors choose compressive tests instead of elongation tests?
Compression testing of hydrogels is common practice and tend provide more accurate data than tensile testing. We have added an additional statement pointing out that the targeted application for the formulated gels is as stationary phase in chromatography columns and thus compression testing is the more relevant test to perform as it resemble a similar load scenario.
A schematic illustration of the chemical structures and the hydrogel formation would be very helpful for making the discussion clear.
We are unsure which parts of the discussion are required to be clarified by an illustration. Given the length of the manuscript we chose to omit any figures that do not provide discussed data.
Reviewer 3 Report
In this manuscript, the authors investigated how the addition of physical-crosslinking agent (cellulose powder) and/or chemical cross-linking (ECH) affect the mechanical properties of cellulose hydrogels. Overall, the experiments are well designed and the obtained results are well explained and supported by the experimental data. The publication of this manuscript will be helpful for other researchers in this field. However, there are some minor problems which need to be rectified before publication.
1. The title of the manuscript is too general. It needs to be improved to better present the content of this manuscript.
2. In Figure 2, the sizes of the data points are too small, making the different shapes difficult to distinguish, especially when the points are closed to each other (the two points on the left).
3. In Figure 3, in the legend, the authors claim: “a single cellulose particle is visible that is closely surrounds by the regenerated, porous cellulose matrix”. However, it is not obvious in the picture. Could the authors label some single cellulose particles in the figure to make this statement more convincing?
4. In the legend of Figure 5 and Figure 7, the authors had better isolate the different groups with “;” or blank space. It can be confusing for some readers with the current format. Besides, in Figure 7, as the line is very thin in the legend, it is very hard to distinguish the green color and blue color.
5. As all the SEM samples are obtained by freeze drying, can the porous structure truly represent the hydrogel structure? Because the pores might be caused by the formation of ice crystals in the matrix.
Author Response
We thank the reviewer very kindly for the critical and insightful comments provided. Please find our point-by-point response below:
1. The title of the manuscript is too general. It needs to be improved to better present the content of this manuscript.
The title has been updated to more adequately describe the work presented.
2. In Figure 2, the sizes of the data points are too small, making the different shapes difficult to distinguish, especially when the points are closed to each other (the two points on the left).
The figure has been updated to hopefully provide more clarity.
3. In Figure 3, in the legend, the authors claim: “a single cellulose particle is visible that is closely surrounds by the regenerated, porous cellulose matrix”. However, it is not obvious in the picture. Could the authors label some single cellulose particles in the figure to make this statement more convincing?
The figure has been updated to hopefully provide more clarity.
4. In the legend of Figure 5 and Figure 7, the authors had better isolate the different groups with “;” or blank space. It can be confusing for some readers with the current format. Besides, in Figure 7, as the line is very thin in the legend, it is very hard to distinguish the green color and blue color.
The figure has been updated to hopefully provide more clarity.
5. As all the SEM samples are obtained by freeze drying, can the porous structure truly represent the hydrogel structure? Because the pores might be caused by the formation of ice crystals in the matrix.
We appreciate the possibility of ice crystal formation during freeze drying which might affect the microstructure of prepared sample. However comparative analysis of microstructures specifically pore size are routinely completed via freeze drying and SEM. Future work may look at the effects of ice crystal formation on pore size either by solvent exchange or temperature gradient. This has been clarified in section 2.8 and corresponding references have been added.
We are also investigating using mercury intrusion porosimetry for future work to analyze the gels in their hydrated state.
Reviewer 4 Report
The article presented addresses a theme of interest regarding the production of cellulosic hydrogels, a classic material, but still having surprising aspects to apply. some observations are to be made:
The title is not entirely suggestive for the content. An example of a title for the study presented: "Physical and Chemical cross-linking cellulose hydrogels with new improved mechanical properties".
In the Introduction, newer bibliographic references should be introduced, especially for the aspects presented in the first two paragraphs of the introduction.
Attention should be paid to uniformizing the resolution and formatting of the figures, and their legend (for example, Figures 4 to 5, Figure 10).
For the bibliographic reference list, new, more recent indices may be added.
Author Response
We thank the reviewer very kindly for the critical and insightful comments provided. Please find our point-by-point response below:
The title is not entirely suggestive for the content. An example of a title for the study presented: "Physical and Chemical cross-linking cellulose hydrogels with new improved mechanical properties".
The title has been updated to more adequately describe the work presented.
In the Introduction, newer bibliographic references should be introduced, especially for the aspects presented in the first two paragraphs of the introduction.
For the bibliographic reference list, new, more recent indices may be added.
More and more recent references have been added to the introduction and throughout the manuscript.
Attention should be paid to uniformizing the resolution and formatting of the figures, and their legend (for example, Figures 4 to 5, Figure 10).
Some figures have been updated to improve clarity. We would kindly ask the reviewer to specify which further changes might be required to improve the figures.
Reviewer 5 Report
Review report
Manuscript ID: materials-468472
Title: Controlling the mechanical properties of cellulose hydrogels
In this article, the authors have studied the effects of different gelling conditions on mechanical properties of prepared cellulose hydrogels. The title of the article seems incomplete and does not sound good. The work is lack of novelty as there is many reported literatures on cellulose based hydrogels and their mechanical properties. The objective of the work is not very clear in the article. Major revisions are necessary before considering the article for publication.
Please find the comments below:
1. The title of the article does not sound good. Please change the title so that one can get an idea about the work.
2. Page 3, Section 1, line 93-101: The authors should rewrite this part describing clearly their main goal and objective behind this work. The authors should explain why this combined effect of physical and chemical crosslinking of cellulose in cellulose hydrogel is so important in the context of research regarding cellulose hydrogel.
3. The authors have used urea and NaOH solution as the solvent for cellulose. I would like to know if the cellulose becomes completely soluble in this solvent. The authors should explain why they used the specific temperature -12°C.
4. In the experimental part, different samples have been prepared. The authors used different gelling temperatures, different heating method and different cellulose loading to prepare hydrogel samples. They also prepared samples with ECH crosslinker. It would be nice if you please make a table including sample names and compositions, as it is difficult to follow the work.
5. Please change the figure 2 with simplified version, as it is difficult to follow. According to the authors, there is a decrease in modulus and strength of the sample (with 50% added cellulose) at higher temperature (prepared in water bath) because of degraded cellulose- how did you confirm about the degradation of cellulose.
6. Page 17, section 3.4, line 481-500: Why there are differences in pore size with different amount of cellulose powder and ECH content- the authors should explain this.
Author Response
We thank the reviewer very kindly for the critical and insightful comments provided. Please find our point-by-point response below:
1. The title of the article does not sound good. Please change the title so that one can get an idea about the work.
The title has been updated to more adequately describe the work presented.
2. Page 3, Section 1, line 93-101: The authors should rewrite this part describing clearly their main goal and objective behind this work. The authors should explain why this combined effect of physical and chemical crosslinking of cellulose in cellulose hydrogel is so important in the context of research regarding cellulose hydrogel.
We have added several sentences to the introduction explaining further the motivation for research and specified potential field of application for the created gels which should help explain the used methods of formulation and analysis.
3. The authors have used urea and NaOH solution as the solvent for cellulose. I would like to know if the cellulose becomes completely soluble in this solvent. The authors should explain why they used the specific temperature -12°C.
While the exact effect of temperature on cellulose dissolution in NaOH based systems is still not fully understood it is now widely accepted is that the dissolution must be performed below 0 °C. For the used solvent with added urea it has been reported that best dissolution was achieved at −12 °C. We would like to refer the reviewer to the excellent recent review by Budtova, Tatiana, and Patrick Navard. "Cellulose in NaOH–water based solvents: a review." Cellulose 23.1 (2016): 5-55, for more information.
4. In the experimental part, different samples have been prepared. The authors used different gelling temperatures, different heating method and different cellulose loading to prepare hydrogel samples. They also prepared samples with ECH crosslinker. It would be nice if you please make a table including sample names and compositions, as it is difficult to follow the work.
A table has been created of all prepared samples and analysis methods and will be provided as supplementary information.
5. Please change the figure 2 with simplified version, as it is difficult to follow. According to the authors, there is a decrease in modulus and strength of the sample (with 50% added cellulose) at higher temperature (prepared in water bath) because of degraded cellulose- how did you confirm about the degradation of cellulose.
The figure has been updated to hopefully provide more clarity. Degradation was confirmed through mechanical testing of the gels. Future work will look at the behaviour in more detail.
6. Page 17, section 3.4, line 481-500: Why there are differences in pore size with different amount of cellulose powder and ECH content- the authors should explain this.
We believe the pore size is affected through the complex interactions and potentially counteracting processes that occur during simultaneous gelation and cross-linking reaction and the corresponding formation of covalent and hydrogen bonds. Due to the high complexity of this process the presented research should not been considered fully conclusive on the issue, but instead we hope will lead to future work exploring this complexity in more detail. The manuscript has been modified to clarify our finding better.
Round 2
Reviewer 1 Report
This manuscript was properly revised and can be accepted for publication in the present form.
Reviewer 2 Report
I suggest acceptance as it is.
Reviewer 5 Report
Second Review
Manuscript ID: materials-468472
Title: Controlling the mechanical properties of cellulose hydrogels
In this manuscript, the authors have clarified all the points, I asked for, in my first review. The article has been improved much and overall presentation is nice.
Now the article is ready for publication.